# Monoclonal Antibodies against SARS-CoV-2 Infection: Results from a Real-Life Study before the Omicron Surge

**DOI:** 10.3390/vaccines10111895

**Published:** 2022-11-10

**Authors:** Riccardo Scotto, Antonio Riccardo Buonomo, Giulia Zumbo, Antonio Di Fusco, Nunzia Esposito, Isabella Di Filippo, Mariano Nobile, Biagio Pinchera, Nicola Schiano Moriello, Riccardo Villari, Ivan Gentile

**Affiliations:** Department of Clinical Medicine and Surgery, Section of Infectious Diseases, University “Federico II” of Naples, Via Sergio Pansini 5, 80128 Naples, Italy

**Keywords:** SARS-CoV-2, monoclonal antibodies, COVID-19, early treatment, real-life, chronic kidney disease, serology, immunodeficiency

## Abstract

Despite the lightning-fast advances in the management of SARS-CoV after 2 years of pandemic, COVID-19 continues to pose a challenge for fragile patients, who could benefit from early administration of monoclonal antibodies (mAbs) to reduce the risk of severe disease progression. We conducted a prospective study to evaluate the effectiveness of mAbs against SARS-CoV-2 among patients at risk for severe disease progression, namely elderly and those with comorbidities, before the omicron variant surge. Patients were treated with either casirivimab/imdevimab, sotrovimab, or bamlanivimab/etesevimab. The rates and risk factors for clinical worsening, hospitalization, ICU admission and death (unfavorable outcomes) were evaluated. A stratified analysis according to the presence of SARS-CoV-2 IgG was also performed. Among 185 included patients, we showed low rates of unfavorable outcomes (9.2%), which were more frequent in patients with chronic kidney disease (aOR: 10.44, 95% CI: 1.73–63.03; *p* < 0.05) and basal D-dimer serum concentrations > 600 ng/mL (aOR 21.74, 95% CI: 1.18–397.70; *p* < 0.05). Patients with negative SARS-CoV-2 serology at baseline showed higher C-reactive protein values compared with patients with positive serology (*p* < 0.05) and a trend toward a higher admission rate to SICU and ICU compared with patients with positive serology. Our results thus showed, in a real-life setting, the efficacy of mAbs against SARS-CoV-2 before an Omicron surge when the available mabs become not effective.

## 1. Introduction

Since the beginning of the COVID-19 pandemic in 2020, plenty of efforts have been spent in the race for a cure against SARS-CoV-2. During the first months of the emergency, clinicians taking care of patients with COVID-19 administered drugs such as hydroxychloroquine, azithromycin, and lopinavir/ritonavir that, later, showed no clinical efficacy [1,2]. The availability of several evidence-based treatments (such as corticosteroids, low-molecular-weight heparin, remdesivir and tocilizumab) radically changed the management of severe COVID-19 [3,4]. Moreover, several vaccines for COVID-19 prevention were approved, including both mRNA and viral vector vaccine. These have shown excellent efficacy and safety profiles [5,6] significantly contributing to reducing the impact of the pandemic in terms of severe disease incidence, hospitalizations and deaths [7].

Despite the unquestionable usefulness of SARS-CoV-2 vaccination, fragile categories of patients may have a sub-optimal response to vaccines. Older patients and those with primary or secondary immunodeficiencies showed an impaired antibody-mediated response after SARS-CoV-2 vaccination, remaining at high risk for severe COVID-19 [8,9]. The optimal clinical management of SARS-CoV-2 infection in these patients is represented by an early diagnosis and treatment able to minimize the risk of progression towards severe disease. In this context, monoclonal antibodies (mAbs) use has been recently implemented for early treatment of frail patients [10]. In Italy, their use has been licensed for patients with risk factors for severe COVID-19 (including older patients, patients with immunodeficiencies and those with chronic comorbidities) in whom COVID-19 was diagnosed in the previous 10 days [11]. Monoclonal antibodies, alone or combined, currently administrable in Italy are: casirivimab/imdevimab, bamlanivimab/etesevimab and sotrovimab. According to a recent meta-analysis, administration of mAbs may reduce the risk of hospitalization, oxygen requirement, invasive mechanical ventilation, and death [12]. Nevertheless, the authors concluded that the evidence of mAbs efficacy is still low especially among non-hospitalized individuals, and that further and long-term studies are needed. In this scenario, we conducted a retrospective, observational real-life study to assess efficacy and safety of mAbs in patients with early mild/moderate disease with the presence of risk factors for progression to severe COVID-19, according to the indication provided by the Italian Drug Agency (AIFA, Agenzia Italiana del Farmaco).

## 2. Methods

This real-life study was conducted among all inpatients and outpatients diagnosed with SARS-CoV-2 infection referred to the Unit of Infectious Diseases at the University of Naples Federico II, Campania Region, Italy, from 1 February 2021 to 6 December 2021 who were treated with anti-SARS-CoV-2 mAbs. The enrolment was stopped on the 6 December 2021 when the first case of the Omicron variant of concern (VoC) of SARS-CoV-2 was confirmed in Italy. No inclusion or exclusion criteria were used, in order to provide real-life results not affected by any selection criteria. In Italy, the administration of mAbs for COVID-19 is regulated by strict indications provided by AIFA [11]. Accordingly, only non-hospitalized adult patients (or adult patients hospitalized for reasons different from COVID-19) who received early treatment (within 10 days from symptoms onset) with mAbs were included. Moreover, early-treatment can be administered to patients who do not require oxygen supplementation and who are at high risk for severe COVID-19 due to older age (>60 years) or comorbidities (e.g., obesity, chronic kidney disease, cardiovascular disease, chronic pulmonary disease, immunodeficiency). All the enrolled patients had to provide a positive molecular oro-rhino-pharyngeal (ORP) swab for SARS-CoV-2 (by RT-PCR) performed in the previous 10 days and were asked to sign an informed consent form on the day of mAbs administration (T0). Moreover, a blood sampling for routinary blood tests (including blood cell count, white cell count, C-reactive protein [CRP], procalcitonin [PCT], lactate dehydrogenases [LDH]) and SARS-CoV-2 IgG dosing, as well as arterial blood gas (ABG) analysis were collected before treatment infusion. Patients’ refusal to perform blood sampling and ABG was not considered an exclusion criterion to reflect the real-life nature of the study. All enrolled patients were treated with either casirivimab/imdevimab 600 mg + 600 mg, sotrovimab 500 mg, or bamlanivimab/etesevimab 700 mg + 1400 mg. The treatment of choice was selected by the medical staff according to local availability. In fact, as the use of mAbs in patients with SARS-CoV-2 infection was initially authorized as an emergency treatment by AIFA, before the final approval of local and international regulatory agencies, the sorting and distribution of limited stocks were managed by a regional crisis unit. Both inpatients and outpatients also performed a follow-up visit at 7 days after mAbs administration (T1). At T1, patients underwent clinical examination, blood test analysis and ABG, and they were asked about the occurrence of adverse drug reactions (ADRs). Only ADRs related to mAbs administration were recorded according to medical judgement. Outpatients were asked to contact the medical staff in case of worsening of symptoms or occurrence of ADRs, whereas inpatients were monitored daily and also asked to inform medical staff in case of worsening ADR if discharged. Outpatients who showed a worsening in clinical conditions were admitted in hospital and regularly performed the T1 follow-up visit 7 days after mAbs administration. The prevalence of the following outcomes was collected: hospitalization (among outpatients), increase of oxygen supplementation, admission in a sub-intensive care unit (SICU), admission in an intensive-care unit (ICU), and death. Increase in oxygen supplementation was defined as the occurrence of desaturation requiring oxygen therapy in patients who previously showed satisfactory saturation percentages at T0, or increased oxygen requirements (as a fraction of inspired oxygen (FiO_2_) or oxygen level) in patients already on oxygen therapy. Admission to SICU, ICU and death were defined as “unfavorable outcomes” for COVID-19. We also defined as ORP viral clearance time the number of days between the first positive test swab for SARS-CoV-2 RNA and the first negative one. This study was conducted according to the world medical association declaration of Helsinki on ethical principles for medical research involving human subjects. The study protocol was approved by the local ethical committee (Prot. N. 88/2022 ID: N.1032)

### Statistical Analysis

All the variables were tested for parametric/non-parametric distribution with the Kolmogorov-Smirnov test. Comparisons between categorical dichotomous variables were performed with the χ^2^ test (or with Fischer’s exact test when applicable), while comparisons between quantitative variables were conducted with the T-student test (parametric variables) or the Mann-Whitney’s U test (non-parametric variables). A descriptive statistical analysis was conducted on clinical and laboratory variables collected at both T0 and T1. For continuous variables, the difference between T1 and T0 were calculated and reported as “delta” (∆). A stratified comparative analysis between T1 and T0 was conducted according to pre-infusion (at T0) serum SARS-CoV-2 IgG status (positive or negative serology). Outcome rates and prevalence of ADRs were reported among the whole study sample and stratified according to baseline serum SARS-CoV-2 IgG status. Finally, univariate and multivariate logistic regression analysis were conducted to perform the risk analysis for the occurrence of at least one unfavorable outcome. Variables associated with at least one unfavorable outcome at the univariate analysis with a *p*-value < 0.2 were included in the multivariate adjusted model. For all the tests, a *p*-value < 0.05 was considered significant. IBM SPSS© version 27 was used for statistical analysis.

## 3. Results

### 3.1. Population Characteristics

According to the AIFA criteria for monoclonal antibodies administration, 185 patients were included in the study (110 outpatients, 75 inpatients). The majority of the patients were female (60.0%), while the median age was 57 years (IQR: 37–72) (Table 1). Most patients (115, 62.2%) had at least one comorbidity among the following: chronic kidney disease, diabetes, immunodeficiency, cardiovascular disease, chronic liver disease, chronic pulmonary disease, neurodegenerative disease, obesity, haemoglobinopathy. In detail, 58 patients (31.4%) had one comorbidity, 31 patients (16.8%) had 2 comorbidities, and 26 patients (14.0%) had ≥3 comorbidities. The most common comorbidities were obesity (22.2%), cardiovascular disease (19.5%) and immunodeficiency (13.0%). Despite the high frequency of comorbidities in the study population, only 55.7% had received SARS-CoV-2 vaccination and only 45.9% of the enrolled patients who performed serum SARS-CoV-2 IgG dosing (163, 88.1% of the total sample) showed a positive result. The most commonly administered mAbs combination was casirivimab/imdevimab (70.3%). At enrollment time (T0), most patients (151, 81.6%) did not require oxygen supplementation and had a median P/F ratio of 462 (IQR: 452–467). Therefore, only 18.4% of patients required oxygen supplementation at enrollment. Notably, at T1 the percentage of patients who required oxygen supplementation therapy decreased to 7.6%. When compared to T0, no clinically significant differences were observed in the laboratory parameters at T1 (Appendix A). Globally, enrolled patients showed favorable outcomes, with a low rate of increased oxygen supplementation (9.7%), SICU admission (4.9%), ICU admission (1.1%) and exitus (4.9%). Seventeen patients (9.2%) showed at least one unfavorable outcome, while 4 out of 110 outpatients (3.6%) needed hospitalization. ADRs occurred in 34 (18.4%) patients. Fever was almost the only ADR reported (33 out of 34 patients 97%), while just one patient had vomiting. In most cases, fever resolved within 24–36 h from its occurrence, with or without the aid of antipyretic drugs.

### 3.2. Factors Associated with Unfavorable Outcome

At the univariate outcome analysis, age > 60 years, CKD, diabetes mellitus, chronic liver disease, chronic pulmonary disease, neurodegenerative disease, obesity, Charlson comorbidity index > 2 and a plasmatic D-dimer > 600 ng/mL were associated with an unfavorable outcome (at least one among SICU, ICU and death) (Table 2). At the multivariate analysis R^2^ = 0.209), patients with CKD (aOR: 10.44, 95% CI: 1.73–63.03; *p* < 0.05) and basal D-dimer serum concentrations > 600 ng/mL (aOR 21.74, 95% CI: 1.18–397.70; *p* < 0.05) were found to be independent risk factors for unfavorable outcome. In particular, 28.6% and 50% of patients, who needed SICU/ICU admission and died, respectively, had CKD. Notably, CRP values, negative SARS-CoV-2 serology at admission and incomplete SARS-CoV-2 vaccination were not associated with an increased risk of unfavorable outcome.

### 3.3. The Possible Role of SARS-CoV-2 Serology

In the stratified analysis between clinical and laboratory parameters at T1 and T0, patients with negative SARS-CoV-2 serology at baseline showed a significant reduction in serum CRP compared with patients with positive serology (median ∆ −16.5 [IQR: −49.2 to −0.4] vs. −1.7 [IQR: −20.0 to +0.4], *p* < 0.05). No other significant differences in clinical or laboratory parameters between T1 and T0 were recorded. Overall, patients with negative SARS-CoV-2 serology had a higher SICU and ICU admission rate compared with patients with positive serology (6.5% vs. 1.2%, *p* = 0.084), as well as a higher rate of ADRs (28.6% vs. 7.1%, *p* < 0.001). No differences in the hospitalization rate, increase in oxygen supplementation or death were recorded (Table 3). Both median ORP viral clearance time (15 days [IQR: 10–20] vs. 14 days [IQR: 9–18; *p* = 0.308]) and median time between positive SARS-CoV-2 RNA on ORP swab and mAbs infusion (3 [IQR: 1–5] vs. 3 [IQR: 1–4]; *p* = 0.717) were similar in patients with negative and positive SARS-CoV-2 IgG serology.

## 4. Discussion

In our study, we showed real-life data on monoclonal antibodies treatment in patients with SARS-CoV-2 infection in a tertiary care center of Southern Italy. The study was conducted when alpha e delta SARS-CoV-2 VoCs were prevalent in Italy and was stopped when the first Omicron VoC case was detected in the country. In this setting, the efficacy of all these mAbs combinations was high beyond any doubt [13]. Noteworthy is that our study population characteristics are quite different from those enrolled in phase 2/3 clinical trials that have led to mAbs authorized use against SARS-CoV-2 infection. In fact, only a minority of the patients enrolled in clinical trials had risk factors for severe COVID-19 such as CKD or immunodeficiencies [14,15], which were relatively common in our study population (CKD: 9.1%, immunodeficiency: 12.8%,). For instance, in the phase 3 study by Weinreich et al., only 1.3% and 3.2% of patients treated with casirivimab/imdevimab had CKD and immunodeficiencies, respectively.

Despite the frailty of our study population, we showed a poor adherence to the vaccination program as only 55% received at least one dose of the anti SARS-CoV-2 vaccine. In this setting, the early treatment with monoclonal antibodies was crucial, also considering that just 45% of the tested patients had IgG positivity to the anti-Spike protein. Nevertheless, the outcome of the disease was generally good. Firstly, we showed an overall improvement in patients’ clinical conditions at T1 compared to T0, with a lower rate of patients requiring oxygen supplementation (7.6 vs. 18.4%) and lower median CRP values. This is particularly significant considering that impaired clinical conditions associated with systemic inflammation are expected after 7–10 days from the diagnosis according to the SARS-CoV-2 infection pathogenesis [16]. Furthermore, our population had relatively low rates of hospitalization, SICU/ICU admission and death. It is plausible that the 10% of included patients with CKD biased these results. In fact, a significant percentage of patients who required hospitalization (20%) or ICU/SICU (22.2%) had CKD, while an even higher percentage (40%) of patients who died had CKD. In multivariate analysis, there was indeed a correlation between CKD and high D-Dimer levels (>600 ng/mL) at T0 and an unfavorable outcome. In detail, patients with CKD showed a 10-fold risk of unfavorable outcome compared with patients without (aOR: 10.44; 95% CI: 1.73–63.03, *p* < 0.05). To our knowledge, no data reporting on the efficacy of monoclonal antibodies in CKD patients are currently available. Since the first wave of COVID-19, it has been demonstrated that CKD patients had a three-fold risk of developing severe COVID-19 compared with patients without CKD [17], while patients with CKD stages 3 to 5 according to KDIGO had a significant increase in mortality rate when compared with those without kidney disease (11.1 vs. 4%) [18]. For this reason, it is well known that CKD is one of the main indications for early treatment of COVID-19. For instance, the RECOVERY trial, in which the efficacy of casirivimab/imdevimab in outpatients with severe COVID-19 was evaluated [19], and the sotrovimab registration trial [15] had none and one patient with CKD respectively. In one of the first real-life studies by Savoldi et al. comparing efficacy of different mAbs combinations, 1.7% of patients were on dialysis for end stage renal disease and no impaired efficacy in this category of patients was shown. It must however be noted that no patients with other stages of CKD, except end-stage disease on dialysis, were enrolled in the study [20].

It is already known that D-Dimer levels are associated with worse outcomes. In 2021, Poudel et al. showed how, in a cohort of 182 patients, a D-Dimer value higher than 1500 ng/mL was an accurate biomarker for mortality, with a sensitivity of 70.6% and a specificity of 78.4% [21]. Our study confirmed the association between high D-dimer values and poor outcome, interestingly within a group of patients in their first stage of the disease at high risk of developing severe COVID-19. In fact, previous data came from patients already hospitalized for COVID-19.

Overall, treatment with mAbs was well tolerated. A treatment-related ADR was reported in 18.4% of patients, with fever being the most common ADR. Even though fever was considered a mAbs-related ADR, it the difficulty in discriminating an actual ADR from a COVID-19 symptom must be underlined, especially considering that all patients were treated within the first days of the disease and might not have already shown fever as part of the COVID-19 related symptom. This consideration may explain the higher rate of ADRs reported in seronegative patients, as it is well-known that vaccinated subjects tend to have milder symptoms if infected.

Lastly, we conducted a stratified analysis based on serum anti-spike IgG, as patients with a negative SARS-CoV-2 serology (either unvaccinated or non-responders to vaccination) were considered at higher risk of clinical worsening. We showed that patients with baseline negative SARS-CoV-2 serology had a significant reduction in serum CRP at T1 compared to patients with baseline positive serology (median ∆ −16.9 [IQR: −51.6 to −0.4] vs. −1.7 [IQR: −20.0 to +0.4], *p* < 0.01). Although in the absence of a control group, a definite statement is still not possible, we can postulate that mAbs treatment may have a high clinical significance in patients with negative anti-spike serology as an increase in CRP and other pro-inflammatory markers at baseline has been widely associated with worse outcome in patients with COVID-19 [22,23]. As mAbs binding to SARS-CoV-2 spike domain leads to inhibition of virus replication and subsequent blockage of the cytokinin cascade triggered by the virus, a reduction in CRP values could be induced by their usage [24].

In our study, despite monoclonal antibodies administration, we showed that patients with negative serology had a higher risk of SICU/ICU admission compared with patients with positive serology. Although we cannot draw final conclusions, we can hypothesize that, as patients with negative serum SARS-CoV-2 spike IgG were those at higher risk of severe progression of COVID-19, mAbs administration could have reduced the rate of unfavorable outcomes. Unfortunately, the lack of a control group did not allow us to evaluate the real efficacy of mAbs drugs in this setting.

## 5. Conclusions

In conclusion, this is one of the first prospective studies supporting the efficacy of monoclonal antibodies in a real-life setting. In a cohort of frail patients, including those with immunosuppression and CKD, we showed a low rate of hospitalization, ICU/SICU admission and death. The lack of a control group is surely a major limitation of our study and did not allow us to directly confirm efficacy of this therapeutic approach in such a population. However, we believe that results from this work are relevant since they shed light on a class of drugs that should be the cornerstone of early treatment of SARS-CoV-2 in the near future.

## Figures and Tables

**Table 1 vaccines-10-01895-t001:** Demographic and clinical characteristics of the enrolled patients (*n* = 185).

Age (Median, IQR)	57 (37–72)
Age > 65 years (*n*, %)	64 (34.6)
Sex (male; *n*, %)	74 (40.0)
Hospitalization regimen (*n*, %)	
- Outpatients	110 (59.5)
- Inpatients	75 (40.5)
Comorbidities	
- Chronic kidney disease ^#^	17 (9.2)
- Diabetes	13 (7.0)
- Immunodeficiency	24 (13.0)
- Cardiovascular disease	36 (19.5)
- Chronic liver disease	6 (3.2)
- Chronic pulmonary disease	20 (10.8)
- Neurodegenerative disease	6 (3.2)
- Obesity	41 (22.2)
Body mass index (median, IQR)	26 (25–30)
Charlson comorbidity index (median, IQR)	2 (0–4)
MASS Score * (median, IQR)	2 (0–4)
Patient vaccined against SARS-CoV-2 (*n*, %)	103 (55.7)
Among these, patient with full course vaccination ^§^	68 (66.0)
SARS-CoV-2 IgG (*n*, %)	
- Positive	85 (45.9)
- Negative	77 (41.6)
- Not Available	23 (12.4)
Monoclonal antibodies administred (*n*, %)	
- Casirivimab-imdevimab	130 (70.3)
- Sotrovimab	16 (8.6)
- Bamlanivimab-etesevimab	39 (21.1)
Time between positive ORP swab and mAbs infusion (days; median, IQR)	2 (1–4)

^#^ Stage 3 to 5 according to KDIGO. * MASS score was calculated according to the US Food and Drug Administration (FDA) Emergency Use Authorization eligibility criteria, as follows: age ≥ 65 (2 points), BMI ≥ 35 (1 point), diabetes (2 points), chronic kidney disease (3 points), cardiovascular disease in patient ≥ 55 years (2 points), chronic respiratory disease in patient ≥ 55 years (2 points), hypertension in patient ≥ 55 years (1 point) and immunocompromised status (3 points). ^§^ second dose or booster dose during the previous 4 months. ORP: oro-rhino-pharyngeal.

**Table 2 vaccines-10-01895-t002:** Univariate and multivariate logistic regression analysis for SARS-CoV-2 infection unfavorable outcome.

	Univariate Analysis			Multivariate Analysis		
	OR	95% CI	*p*-Value	aOR	95% CI	*p*-Value
Male Sex	1.78	0.65–4.85	0.258	-	-	-
Age > 60 years	6.09	1.69–22.00	<0.01	0.63	0.01–106.11	0.858
Comorbidities						
- CKD *	10.64	3.13–36.11	<0.001	10.44	1.73–63.03	<0.05
- Diabetes	4.38	1.03–17.85	<0.05	4.78	0.49–46.57	0.178
- Immunodeficiency	1.9	0.49–7.36	0.355	-	-	-
- Cardiovascular disease	1.69	0.50–5.73	0.401	-	-	-
- Chronic liver disease	6.79	1.13–40.90	<0.05	0.7	0.01–33.48	0.855
- Chronic pulmonary disease	5.63	1.67–19.98	<0.01	4.17	0.65–28.83	0.133
- Neurodegenerative disease	2.49	0.27–22.95	0.42	-	-	-
- Obesity	4.45	1.32–14.92	<0.05	4.62	0.89–23.89	0.068
Charlson comorbidity index > 2	3.2	1.08–9.49	<0.05	3.81	0.02–707.71	0.615
Incomplete SARS-CoV-2 vaccination schedule	1.14	0.21–6.33	0.876	-	-	-
Negative SARS-CoV-2 IgG	1.35	0.37–4.62	0.63	-	-	-
Laboratory parameters at admission						
- Lymphocyte count < 1000 cell/µL	2.25	0.82–6.15	0.114	2.42	0.49–12.17	0.28
- D-dimer > 600 ng/mL	3.5	0.96–12.75	0.058	21.74	1.18–397.97	<0.05
- CRP > 60 mg/L	1.53	0.51–4.62	0.453	-	-	-
- LDH > 300 U/L	1.71	0.56–5.18	0.347	-	-	-
Time between positive ORP swab and mAbs infusion > 5 days	0.81	0.18–3.90	0.817	-	-	-

* Stage 3 to 5 according to KDIGO. OR: odds ratio; 95% CI: 95% confidence interval; CKD: chronic kidney disease; CRP: C-reactive protein; LDH: lactate dehydrogenase; ORP: oro-rhino-pharyngeal.

**Table 3 vaccines-10-01895-t003:** Stratified outcomes analysis according to baseline SARS-CoV-2 serology (and = 162).

	Positive SARS-CoV-2 IgG (*n* = 85)	Negative SARS-CoV-2 IgG (*n* = 77)	*p*-Value
Hospitalization needed (*n*, %) ^#^	3 (4.6)	1 (3.0)	0.586
Increase in oxygen therapy (*n*, %)	8 (9.4)	9 (11.7)	0.637
Exitus (*n*, %)	4 (4.7)	2 (2.6)	0.389
SICU/ICU (*n*, %)	1 (1.2)	5 (6.5)	0.084
ORP viral clearance time (days; median, IQR)	14 (9–18)	15 (10–20)	0.308
ADRs (*n*, %)	6 (7.1)	22 (28.6)	<0.001

^#^ Among 110 outpatients. SICU: sub-intensive care unit. ICU: intensive care unit. ORP: oro-rhino-pharyngeal. ADRs: adverse drug reactions.

## Data Availability

ARB is responsible for data storage.

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
