# Peer review of "Monoclonal Antibodies against SARS-CoV-2 Infection: Results from a Real-Life Study before the Omicron Surge"

_vaccines, 2022, doi:10.3390/vaccines10111895_

Round 1

Reviewer 1 Report

This is a valuable paper and should be published. However it should also be readable, and at present it is not.

Pages 3-6 have the Results. These are presented without sub-headings and hence without a logical flow. They also repeat the data presented in the Tables…further obscuring the message.

Figure 1 lacks a meaningful legend…and I cant follow it…percentages of what?

Table 1 is a little easier to follow but requires formatting. What is TNF?

Table 2: is this an unfavorable outcome from covid or an unfavourable outcome from the administration of MoAb? If the former then what does it tell us about moAb? The Table needs formatting as the lines in columns 1 and 2 do not align. I am surprised that incomplete immunizations has so little consequence.

Finally the multivariable model does not appear to be optimized, and the strength of the model is not provided.

Table 3. Do the p values relate to the delta changes? What time elapsed between T0 and T1? Do we need all these data (eg; WBC and lymphocytes)?

Table 4: “Negativization” of what? I cant relate this to the effect of MoAbs

The work needs careful editing.

Reviewer 2 Report

There are only some minor editing suggestions.

1. To make reading smoothly and more understandable, the Result section with one figure and 4 tables should be divided into several subsections with subtitles.

2. Some paragraphs in the paper are too long. The entire Introduction is just one long paragraph, same as Methods section and the first paragraph of Introduction. It would be better to divide these paragraphs into several ones, for the same reason as above.

3. Line 36-37: “Our results thus showed, in a real-life setting, the efficacy of mAbs against 36 SARS-CoV-2 before Omicron surge when the available mabs become not effective.” Check this sentence. It is not clear about “the available mabs.”

4. More editing is needed. Here are a few of examples, but the entire manuscript should be reread.

line 27-28: “Patients were treated with either casirivimab/imdevimab, sotrovimab, and bamlanivimab/etesevimab.” Also a similar one in line 90-91. (Patients were treated with either casirivimab/imdevimab, sotrovimab, or bamlanivimab/etesevimab.)?

In line 104, “…the regional crisis unit. . All inpatients…”.

Line 171: “… was tendentially associated with unfavorable outcome At the multivariate analysis,…”.

Line 167: “…60 years, CKD, diabetes mellitus,…” (Chronic kindney disease (CKD).)

Line 175-177: “Moreover, 77.8% and 87.5% of patients who needed SICU/ICU and died, respectively had serum D-dimer concentrations > 600 ng/ml at admission, needed SICU/ICU admission and died, respectively.”

Line 258-259: “It must however be said that all included patients were on dialysis an no patients with other stages of CKD were enrolled (28).”

Line 300: “ICU/SICU admission 300 and death. . The lack of a control group…”

Round 2

Reviewer 1 Report

This version is difficult to read because of the tracked changes. The coverletter also has tracked changes and does not specify where the changes are made to the manuscript (page x, line y). R2 should be more readable.

R2 should also be shorter. All information in Table 1 and Figure 1 is in the text. Duplication should be removed….so perhaps delete both. The text can also be shortened and written more clearly. You cite that 6.6% and 9.7% needed oxygen supplementation after treatment. Which is correct?
The n value should be defined when ut is used to calculate a statistic

The headings inserted into the Results section are not informative. A useful heading summarises the findings described under it.

Items unrelated to MAb treatment should be omitted…eg: associations with CRP levels.

Table 2 needs formatting so that the items in column 1 can be aligned with column 2. I THINK that all of the statistics for  the multivariable analyses are derived from a single model (including all factors achieving p<0.2). The strength of this model should be cited.

It would be helpful to know if the factors identified are different from those seen without mAbs….but I accept that this may not be possible.

Tables 3 and 4 should be omitted. Analyses based on delta values are rarely helpful as the values calculated for the range are large relative to the values themselves.

“negativization” is not an English word.

The Introduction and reference list should be reduced by half – retaining only a focus on mAbs.

Round 3

Reviewer 1 Report

The manuscript is certainly improved. However making it readable has exposed the logical hole in the construction of the study. Specifically: the analyses were not designed to answer the question posed in the title – are MAb treatments safe and effective?. A clear illustration of this is the complete absence of mAbs from Table 2, An outcome of “adverse effects “ is logical. However you have tested everything except the MAbs!
Similarly the new Table 3 addresses the effect of pre-treatment Ab levels…which is interesting. However it doesn't directly inform the reader re the effect of mAb. This would require information re the MAb administered ….does it mimic seropositivity?
I also note that the Discussion has not been substantially edited. It contains novel data related to the patients recovery. The data should be omitted if it does not address the issue of the manuscript…or tabulated concisely if it does.
